# Simultaneous Magmatic and Hydrothermal Regimes in Alta–Little Cottonwood Stocks, Utah, USA, Recorded Using Multiphase U-Pb Petrochronology

**Michael A. Stearns [1],\***, **John M. Bartley [2]**, **John R. Bowman [2]**, **Clayton W. Forster [1]**, **Carl J. Beno [2]**, **Daniel D. Riddle [2]**, **Samuel J. Callis [2]** and **Nicholas D. Udy [1]**

1   Earth Science Department, Utah Valley University, Orem, UT 84058, USA; cforster2110@gmail.com (C.W.F.); nicholas.udy@uvu.edu (N.D.U.)
2   Department of Geology and Geophysics, University of Utah, Salt Lake City, Ut 84112, USA; john.bartley@utah.edu (J.M.B.); john.bowman@utah.edu (J.R.B.); carl.beno@utah.edu (C.J.B.); daniel.riddle@gmail.com (D.D.R.); samueljcallis@gmail.com (S.J.C.)
\*   Correspondence: mstearns@uvu.edu

**Abstract:** Magmatic and hydrothermal systems are intimately linked, significantly overlapping through time but persisting in different parts of a system. New preliminary U-Pb and trace element petrochronology from zircon and titanite demonstrate the protracted and episodic record of magmatic and hydrothermal processes in the Alta stock–Little Cottonwood stock plutonic and volcanic system. This system spans the upper ~11.5 km of the crust and includes a large composite pluton (e.g., Little Cottonwood stock), dike-like conduit (e.g., Alta stock), and surficial volcanic edifices (East Traverse and Park City volcanic units). A temperature–time path for the system was constructed using U-Pb and tetravalent cation thermometry to establish a record of >10 Myr of pluton emplacement, magma transport, volcanic eruption, and coeval hydrothermal circulation. Zircons from the Alta and Little Cottonwood stocks recorded a single population of apparent temperatures of ~625 ± 35 °C, while titanite apparent temperatures formed two distinct populations interpreted as magmatic (~725 ± 50 °C) and hydrothermal (~575 ± 50 °C). The spatial and temporal variations required episodic magma input, which overlapped in time with hydrothermal fluid flow in the structurally higher portions of the system. The hydrothermal system was itself episodic and migrated within the margin of the Alta stock and its aureole through time, and eventually focused at the contact of the Alta stock. First-order estimates of magma flux in this system suggest that the volcanic flux was 2–5× higher than the intrusive magma accumulation rate throughout its lifespan, consistent with intrusive volcanic systems around the world.

**Keywords:** incremental pluton emplacement; contact metamorphism; petrochronology; titanite; zircon; U-Pb dating; thermometry; hydrothermal fluids

## 1. Introduction

Transport of magma within the Earth's crust has far-reaching effects: pluton emplacement at a variety of structural levels, development of associated contact and hydrothermal metamorphism in and around the magma conduits (plumbing systems), and commonly, but not always, volcanic eruptions. The relative timing of magma and/or fluid transport in the system is important for understanding mass and heat transfer, but also for volcanic–plutonic connections and volcanic hazard mitigation. The rock record at any one location depends on the interplay between the different processes and parts of the system. The magma emplacement rate and duration [1], tectonic forces [2], pre-existing and evolving permeability structure [3], and metamorphic mineral reactions [4] all serve to modify

and cause feedbacks within the system. Simplifying assumptions, such as an instantaneous magma emplacement (e.g., Annen [5] and Reverdatto et al. [6]) or a lack of magmato-metamorphic processes (e.g., Paterson et al. [7]) exclude critical aspects of mass and heat transfer and obscure our understanding of the integrated system. The term "magmato-metamorphic" is used here to encompass a variety of near- and sub-solidus processes, such as melt and/or magmatic water flow, major and accessory phase (re)crystallization, and mineral reactions, that continue to modify the rock and obscure the record of super-solidus processes. More recently, authors have been applying sophisticated modeling that accounts for incremental pluton emplacement (e.g., Annen [8]) and pulsed conduit flow (e.g., Floess and Baumgartner [9]) to better understand the complex thermochemical feedback between magmas and wall rocks. This paper presents new preliminary petrochronology data from the Alta stock-Little Cottonwood stock system, which demonstrates the complex spatial and temporal patterns of magmatism, hydrothermal infiltration, and contact metamorphism over a crustal section from the surface to an ~11.5 km paleodepth [10].

The Wasatch Intrusive Belt comprises a series of Eocene–Oligocene plutons that intruded the thickened crust in northern Utah following the Sevier orogeny [11–13] (Figure 1A). The Wasatch Intrusive Belt magmas likely resulted from lower-crustal metasomatism and melting caused by the last stages of subduction of the Farallon plate [12,14]. The emplacement and space-making involved extension driven by a combination of far-field tectonic and gravitational forces acting on the thickened crust prior to the Basin and Range extension [15–17]. The plutons intrude a sequence of siliciclastic and carbonate rocks ranging in age from Proterozoic to Triassic. Following emplacement, Wasatch Intrusive Belt plutons were exhumed by range-bounding normal faults of the Oquirrh and Wasatch ranges (Figure 1B) beginning at ~18 Ma (17.6 ± 06 Ma K-Ar sericite date [18] and are still being actively exhumed (e.g., References [19–21]). Offset across the Wasatch fault has tilted the range by ~20° to the east along a roughly horizontal N–S rotation axis [22]. This rotation and exhumation exposed a crustal section from ~11.5 km at the southeast corner of the Salt Lake Valley [10] to the paleosurface (Figure 1C) east of Park City, Utah. From structurally deepest to shallowest, this section contains the Little Cottonwood & Ferguson stocks (included in the Little Cottonwood stock from here forward), Alta stock, Clayton Peak and other eastern stocks, and the Keetley volcanic deposits. Previous geochronology from the Wasatch Intrusive Belt includes a combination of multigrain thermal ionization mass spectrometry U-Pb zircon; K-Ar mica and amphibole; fission-track titanite and zircon; and (U-Th)/He and fission-track in titanite, zircon, and apatite dates [19,20,23–25]. These data suggested that pluton emplacement began at ~36 Ma with the eastern stocks, continued with the emplacement of the Alta stock at ~35 Ma, and ended with the Little Cottonwood stock at ~30 Ma. New petrochronology presented here establishes magmatism in the Little Cottonwood and Alta stocks from ~36–26 Ma, beginning and ending with different portions of the Little Cottonwood stock, thus indicating a different sequence and a much longer duration of pluton emplacement from that previously inferred.

The Alta stock is a structurally shallow (~5–5.5-km depth current exposure [22,26,27] dike-like intrusion with subvertical walls (Figure 1)). The two mappable intrusive phases that comprise the Alta stock, namely an equigranular border phase and a later porphyritic central phase, are distinguished by crystal size, microscopic rock texture, and locally recognizable cross-cutting relationships [28]. The major minerals of the Alta stock are plagioclase + K-feldspar + quartz + biotite + amphibole, with titanite + apatite + zircon + ilmenite ± magnetite accessory phases. The contact metamorphic aureole surrounding the Alta stock is especially well-studied [28–34]. Past and continuing work has largely been motivated by the apparent mismatch between the size of the intrusion (~2 km at its widest) and the surrounding contact aureole (≤1.25 km in width). Cook and Bowman [30,31] demonstrated that metamorphism of the carbonate wall rocks was largely driven by the infiltration of high-temperature (~625 °C), $H_2O$-rich fluids laterally outward from the border phase of the Alta stock, which exploited and modified the existing permeability structure surrounding the pluton (Figures 1 and 2). Numerical models assuming instantaneous emplacement of the pluton showed that the locations of prograde metamorphic isograds, isotherms based on calcite-dolomite thermometry,

and the pattern of $^{18}O/^{16}O$ depletion across the aureole could be duplicated with ~5000 years of advective heat and fluid flow, followed by ~20,000−30,000 years of conductive heating to produce the outer aureole [33]. This numerical modeling indicates that the observed extent of the contact aureole could be matched with heat supplied solely from cooling of the Alta stock only if the duration of pluton emplacement was <5000 years.

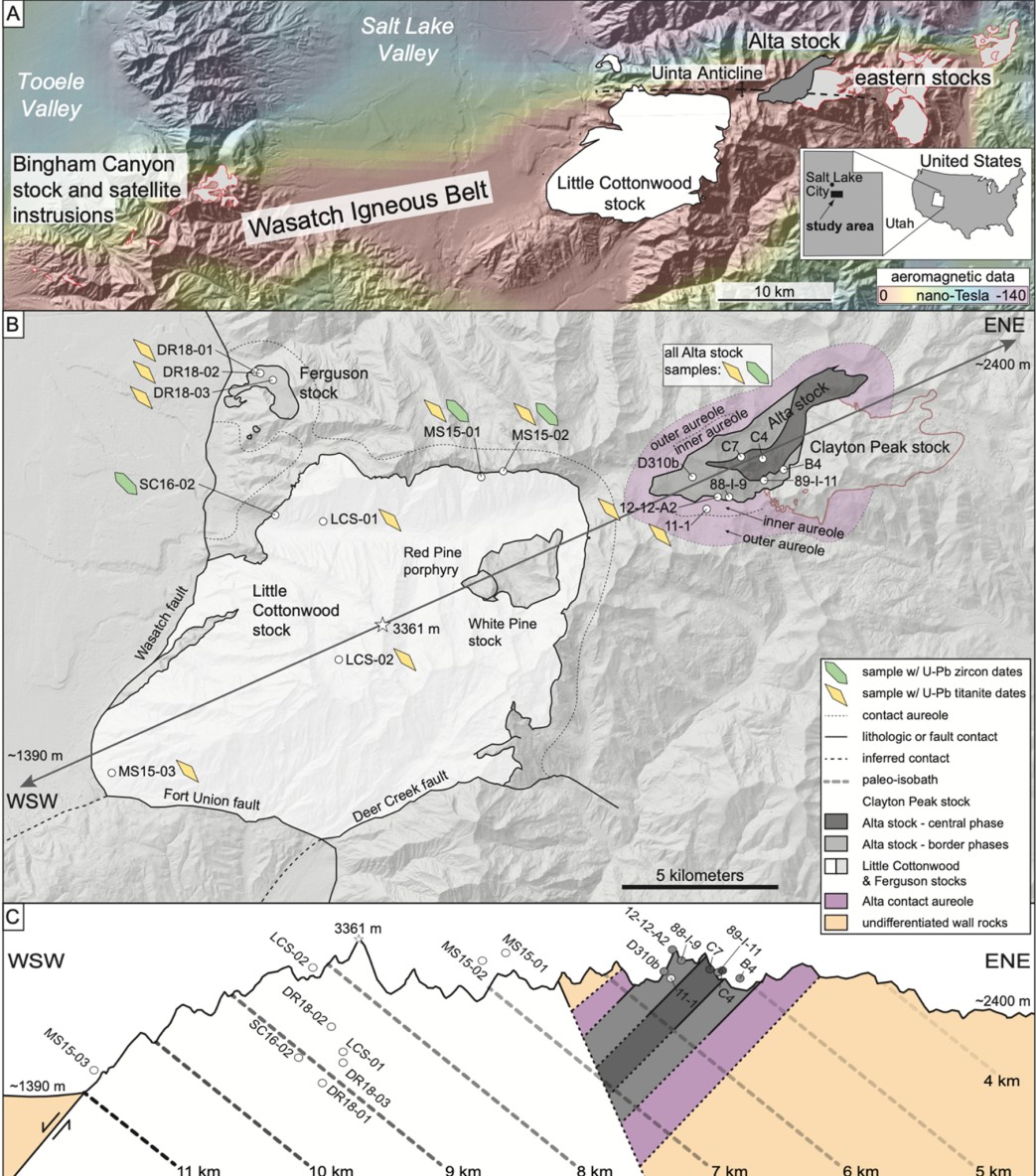

**Figure 1.** (**A**) A simplified geologic map (modified from Wohlers and Baumgartner [35]) shows the location of the Wasatch Intrusive Belt in northern Utah; the correlation of the Wasatch Intrusive Belt with the Uinta Anticline and the aeromagnetic anomaly (nano-Tesla scale bar) that likely represents a continuous body of intrusive rocks at depth [36]. (**B**) The more detailed geologic map overlaid on a shaded relief map and accompanying schematic cross-section (**C**) that illustrates the sample locations and the estimated paleodepth (isobaths) of the analyzed samples [10,22,37]. The cross-section is highly vertically exaggerated (see noted elevations on the map and cross-section) and was calculated using a ~15° rotation due to the differing azimuth from John's [22] ~20° rotation of the Wasatch footwall block. The western Alta and eastern Little Cottonwood contacts are ~75° to WSW and ~50° to ENE, respectively, based on three-point solutions.

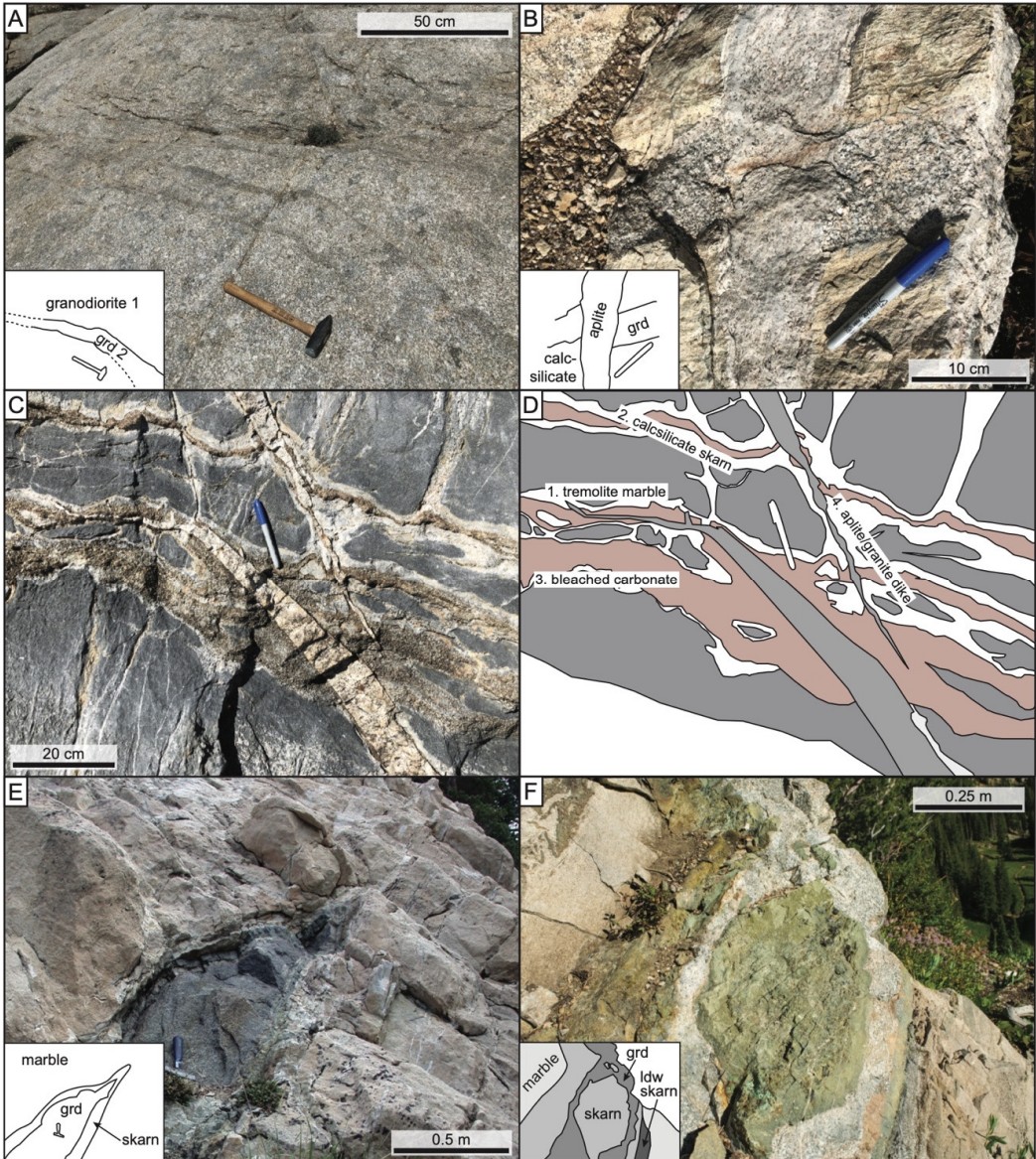

**Figure 2.** Photographs and interpretation schematics showing the highly composite nature of the magmatic and hydrothermal system. (**A**) A granodiorite (grd in schematics **A**, **B**, **E**, and **F**) dike with slightly more mafic margins cross-cutting nearly identical granodiorite in the Little Cottonwood stock. (**B**) A granodiorite dike that intruded the wall rock and was later crosscut by an aplite dike near the southeastern margin of the Little Cottonwood stock. (**C**) A photograph and (**D**) interpretation of a highly composite outcrop that reflects the injection of both magma and aqueous fluid into carbonate wall rock near the southeastern margin of the Little Cottonwood stock. (**E,F**) Calcite marbles intruded by granodiorite sills with calcsilicate skarns developed at the contacts. Calcsilicate skarns are also found adjacent to and included in the Alta stock. Ldw = ludwigite, resulting from boron metasomatism in the Alta aureole [69].

In contrast, if space for the Alta stock was made by horizontal dilation perpendicular to the long axis of the pluton [12] at a reasonable tectonic rate for a continental lithosphere (~1–2 mm·yr$^{-1}$), emplacement of the Alta stock must have taken ~1–2 Myr [38]. Thus, the preferred emplacement model based on field observations implies that the Alta stock grew too slowly by nearly three orders of magnitude to account for the thermal aureole that surrounds it. This extreme mismatch poses two important questions: What is the thermal history of the system, including the Alta conduit, aureole,

and associated Little Cottonwood stock? What are the sources of heat and fluids, beyond the Alta stock, that drove metamorphism in the Alta aureole?

Because the contact aureole is centered on the Alta stock, it is unsurprising that previous studies have assumed that the Alta stock was the source of the heat and fluids that produced the aureole. However, a protracted emplacement of the Alta stock, coupled with the recognition that fluid infiltration is required to drive metamorphic reactions and $^{18}O/^{16}O$ depletion in the aureole, opens up the possibility that at least some of the hot fluid that emanated from the Alta stock originated from another source. The structurally deeper (current exposures correspond to a paleodepth range of ~6.5–11.5 km) and much larger Little Cottonwood stock is bounded by the Wasatch and Deer Creek faults to the west and south, respectively (Figure 1). Data from this study confirmed that the Ferguson stock, interpreted as a satellite to the main Little Cottonwood stock body, is cogenetic. In general, the Little Cottonwood stock is more felsic than the Alta stock, is structurally composite, and its modal mineralogy, crystal size distribution, and chemistry are highly variable [39,40] (Figure 2). The major minerals that make up the Little Cottonwood stock are plagioclase + K-feldspar + quartz + biotite ± amphibole, with titanite + apatite + zircon + ilmenite ± magnetite accessory phases (Figure 3). The Little Cottonwood stock intruded the Neoproterozoic Big Cottonwood Formation on the north and contains a ~1 km × ~200 m screen of these rocks near the western interior of the intrusion (Figure 1). The Little Cottonwood stock magmas intruded Cambrian siliciclastic rocks and Mississippian carbonates on its southeastern margin [41]. Emplacement of the Little Cottonwood stock contact metamorphosed and locally melted the surrounding wall rocks. The Big Cottonwood Formation is composed of interbedded quartzite and shale, and pelitic layers contain the peak contact metamorphic assemblage of biotite + cordierite + sillimanite + K-feldspar + melt [35].

The mineral textures (sensu lato) of intrusive igneous rocks are a topic of active conversation, and a growing body of both field (e.g., Johnson and Glazner [42]) and experimental (e.g., Lundstrom [43]) data suggest that both major and accessory phases in granitic rocks are susceptible to recrystallization at temperatures lower than traditional solidi of ~700 ± 50 °C [44]. For example, the coarsening of orthoclase to so-called megacrysts that are common in calc-alkaline intrusions, including the Little Cottonwood stock, may involve melt (e.g., Higgins [45]) or may occur completely in the solid state [46]. Cathodoluminescence imaging, titanium-in-quartz thermometry in the Alta stock [47], and numerical diffusion modeling of similar data from the Tuolumne Intrusive Suite [48] suggest that quartz in granitic plutons has commonly recrystallized at temperatures in the hydrothermal regime well below traditional granitic solidi. Accessory phases, such as titanite, have been shown to be reactive and recrystallized at similar thermal and fluid conditions [49–51]. Proper interpretation of such overprinting and continued modification using a range of processes spanning magmatic (e.g., Bartley et al. [52]) to hydrothermal (e.g., Smirnov [53]) requires a robust, multifaceted petrochronology approach to relate the chemistry and the time for the process to occur.

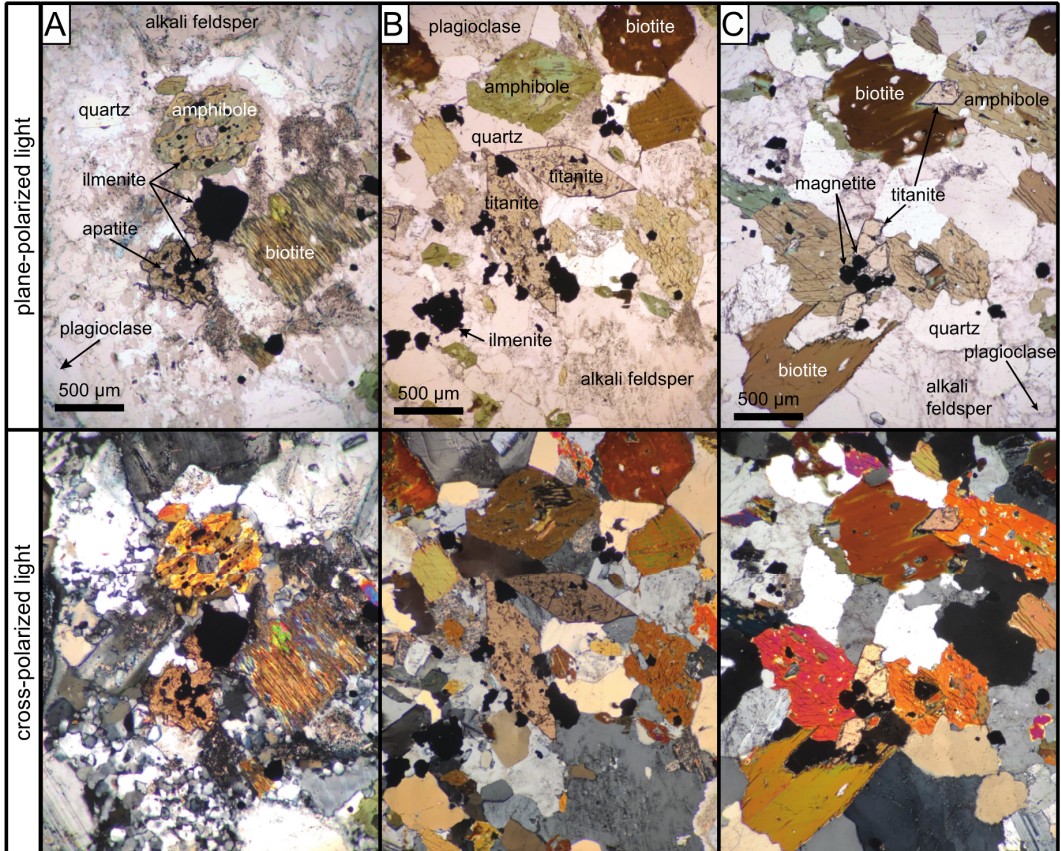

**Figure 3.** Plane polarized (top) and cross-polarized (bottom) photomicrographs that illustrate the range of titanite morphologies and phase relationships observed in both the Alta (**A**,**B**) and Little Cottonwood stocks (**C**). Titanites range from euhedral with few inclusions to anhedral rims on oxide phases (typically ilmenite). Zircons (not seen here) are typically inclusion free and included in a variety of early crystallizing phases, such as plagioclase and biotite. We interpreted these different groups to represent assemblages that grew via different processes. (**A**) Alta central phase sample C4, (**B**) Alta border phase sample D310b, and (**C**) Little Cottonwood stock sample MS15-03. All images were taken at 4× objective magnification.

Petrochronology exploits the accuracy and spatial context of in situ laser ablation split stream (LASS) inductively coupled plasma mass spectrometry (ICP-MS) analysis to simultaneously collect U-Pb isotopes and trace element contents in a variety of chronometer phases [54–56]. Measuring U-Pb and tetravalent cations in multiple phases, such as zircon and titanite from the same rock, gives more complete information about the thermal history of the rock due to the different reactions responsible for the paragenesis of each phase. Both zircon and titanite are common in calc-alkaline rocks like the Wasatch Intrusive Belt (WIB) (Figure 3) [57,58]. Zircon saturation depends on the Zr content and the network modifier/former ratio M of the magma [59,60]. In the Alta–Little Cottonwood system, whole-rock chemistry predicts the zircon saturation at ~725 ± 25 °C. Hanson [39] documented the occurrence of inherited zircon based on petrographic observations, which prolonged the Zr saturation in both the Little Cottonwood and Alta stocks. Prolonged zircon saturation suggests that the zircon saturation temperature ($T_{Zr}$) would be a maximum emplacement temperature for the pluton [39,61]. Titanite paragenesis in calc-alkaline rocks is more complex, and as mentioned above, titanite (re)crystallizes via multiple processes, such as de novo crystallization from melt [62], redox reactions involving Fe-Ti oxides [58], and/or (re)crystallization in the presence of fluids [63]. Titanite is stable in calc-silicate skarns [64–66] and often grows due to

the breakdown of Ti-bearing clinopyroxene, and may (re)crystallize during the infiltration of fluorine and/or $H_2O$-rich fluids [50,67,68].

This paper presents titanite and zircon U-Pb dates, trace element concentrations, and tetravalent cation thermometry from 17 samples in the Alta–Little Cottonwood intrusive–hydrothermal system, which spans the upper ~11.5 km of the Eocene–Oligocene crust in what is now Utah. The new data indicate significant differences from previous conclusions regarding the timing, duration, and spatial extent of magma emplacement and hydrothermal activity within the Alta–Little Cottonwood system. The data together define a temperature–time path that spans >10 Myr from ~36–23 Ma. They further indicate that magmatic and hydrothermal processes were episodic at any given location but they were active in some portion of the system continuously throughout the >10 Myr duration.

## 2. Materials and Methods

Samples were taken from the freshest or least weathered outcrops and trimmed to remove any stained or visually altered rock before cutting thin section billets. As much as possible, titanite and zircon crystals were imaged and inspected for internal chemical zoning prior to analysis using backscattered electron and cathodoluminescence detectors, respectively. These images guided the laser spot placements to minimize the mechanical mixing of chemically and/or isotopically distinct domains and to ensure an analysis of the full range of textural populations in each sample. Many of the zircons analyzed, especially in the Alta stock samples, were chemically zoned at a scale smaller than the analytical beam (~20–25 μm diameter for zircon analyses), and thus the dates were certainly mechanically mixed during some analyses and represent minimum durations of zircon growth.

The majority of samples (except LCS-01,02 and MS15-01,02) were analyzed in thin sections to maintain the petrologic context of the U-Pb dates and trace-element analyses via the laser ablation split stream (LASS) technique; see the method and instrument details of References [49,54,70]. The crystals were ablated using a Photon Machines/Teledyne 193 nm Excimer laser equipped with a two-volume Helex® stage [71] and the ablated material was introduced via He carrier gas to the multi-collector inductively-coupled plasma (ICP-MS) and quadrupole (Q-MS) mass spectrometers to measure uranium and lead isotopes and trace elements, respectively. Isotopic ratios were measured on a Nu Plasma 3 and Thermo NeptunePlus multicollector ICP-MS and trace elements were measured on Agilent 7500ce and 8900 quadrupole mass spectrometers. A laser spot diameter of 20–40 μm for titanite and 20–25 μm for zircon, and a laser repetition rate of 4–6 Hz and fluence of ~2–3 J/cm$^2$ were used during the in situ analyses.

Isotopic analyses were bracketed using and standardized to a matrix-matched primary isotopic reference material (91500 zircon, Bear Lake Road and MKED-1 titanites [72–76]). Matrix-matched secondary reference materials were analyzed during all the zircon analyses (Plesovice zircon [77]) and titanite analyses from twelve samples (BLR titanite) to determine the accuracy and propagate external uncertainties. During the titanite analyses from six samples, both "Yates Mine" titanite [70] and Plesovice zircon (standardized to 91500 primary reference material) were used as a secondary reference material due to a lack of titanite standards at the time (see the Data Repository File).

Mass spectrometry data were reduced and interpreted using the VizualAge Data Reduction Scheme [78] for the Iolite plugin within IgorPro [79]. Homogeneous portions of the time-resolved data were selected to minimize the common Pb by monitoring the $^{204}$Pb and $^{208}$Pb channels and maximize the concordance. The isotopic dates were not used as a criteria for selecting portions of the data. The Iolite output data were further reduced using the Isoplot plugin for Excel [80] and the IsoplotR package for the R statistical program [81]. Analytical uncertainty, counting statistics, and extra error for the homogeneity and accuracy of the secondary reference materials were propagated in quadrature for the final reported uncertainty (see the Data Repository File), which was typically ~2–3% or ~0.5–1.0 Ma for these samples. The typically high variance and discordance of U-Pb in titanite data, and some in situ zircon data, require an interpretation of the raw dates using a partial Pb isochron method [82]. Initial $^{207}$Pb/$^{206}$Pb values were chosen using a combination of linear regression, visual data fitting,

calculation using Stacey–Kramer Pb growth curves, and comparison to Pb isotope data from K-feldspar. All of the initial Pb values used to calculate the [207]Pb-corrected dates fell within the expected range of 0.84–0.87, and the uncertainty in the initial Pb isotope ratio was propagated into the reported individual dates from each analysis point.

Ti-in-zircon and Zr-in-titanite apparent temperatures were calculated using the Ferry and Watson [83] and Hayden et al. [84] calibrations, respectively. Both calibrations depend on the activities of $TiO_2$ and $SiO_2$. The activity of $TiO_2$ was chosen to be $\alpha_{TiO2} = 0.5 \pm 0.2$ for both thermometers based on three observations: (1) titanite was present and rutile was absent in all samples except SC-02, which also lacked titanite ($\alpha_{TiO2} \geq 0.5$ except SC-02 [84]); (2) non-exsolved Fe-Ti oxide pairs from the cogenetic WIB extrusive rocks produced $\alpha_{TiO2} \sim 0.5$–0.7 [39,85]; and (3) the rhyolite-MELTS model matching the modal abundance of major phases (for the Alta stock–Little Cottonwood stock (AS–LCS) range of whole-rock chemistry) also produced $\alpha_{TiO2} \sim 0.5$–0.7 [86]. For the Ti-in-zircon thermometer, the reported uncertainty of ~±35 °C includes the analytical uncertainty (~10% 2SE), uncertainties in the activity terms (~0%–40%), and P dependence (~10%). The plotted apparent temperatures include the estimated pressure dependence of 35 °C/GPa. The Data Repository File includes temperatures with and without the pressure corrections calculated using both the lower [83] and the higher pressure dependence of Ferriss et al. [87] of 100 °C/GPa. The Zr-in-titanite thermometer is much more dependent on pressure and the typical uncertainty is ±40–50 °C.

## 3. Results

### 3.1. Petrologic Observations

The Alta and Little Cottonwood stocks are texturally [28,40] and compositionally variable [12]. Internal intrusive contacts were locally visible and suggest that the stocks are also highly composite. Granitic and aplite dikes were common both within and outside the intrusions (Figure 2A,B). Additionally, wall rocks adjacent to both intrusions recorded many generations of melt and fluid infiltration (Figure 2C–F). Where dikes intruded the wall rocks at the margins of the Alta and Little Cottonwood stocks, the dikes were commonly deformed and suggested syn-deformation emplacement (Figure 2C,D).

At the thin section scale, titanite in both intrusions ranged from euhedral with no inclusions to anhedral and overgrowing ilmenite (Figure 3). Titanite was commonly associated with both ilmenite and magnetite, and with amphibole, biotite, apatite, and sometimes plagioclase. This assemblage suggests a titanite-forming reaction similar to that seen in amphibolite. Titanite was commonly included within quartz, alkali feldspar, and amphibole (Figure 3), but titanite included within plagioclase was not observed. Titanites in both intrusions contained internal chemical zoning, including oscillatory zoning, sector zoning, and patchy zoning indicative of a range of processes from neocrystallization from a granitic melt, neocrystallization from pre-existing phases, and fluid-aided interface-coupled dissolution-reprecipitation recrystallization [51]. Zircons were typically euhedral and were observed to be included within most major minerals, including plagioclase (Figure 3A). Zircons ranged from nearly equant to acicular and oscillatory zoning was common. Zircons in the Little Cottonwood stock often contained distinct mantles and rims that were typically cathodoluminescent dark and bright, respectively. Resorbed, inherited cores were observed and are interpreted to have traveled from the melt source and served as nuclei for further zircon growth (Figure 4).

### 3.2. Petrochronology

#### 3.2.1. U-Pb Dates

We presently have many more isotopic dates from titanite (n = 17 samples, 766 analyses) than from zircon (n = 9 samples, 254 analyses), particularly in the Little Cottonwood stock (titanite: n = 8 samples, 326 analyses; zircon: n = 3 samples, 129 analyses) (Figure S1 and Table S1). Isotopic dates are

reported as $^{207}$Pb-corrected dates projected to concordia from an initial $^{207}$Pb/$^{206}$Pb ratio for each sample (Data Repository). For discussion of this method, see the Materials and Methods Section above. The mean square weighted deviations (MSWD [88]) of the Little Cottonwood stock zircon date populations ranged from ~6 to 21. Dates from three samples (MS15-01, 02, and SC16-02) ranged from 34–26 Ma, 35–25 Ma, and 33–27 Ma, respectively, with Proterozoic (~1000 Ma) inherited cores. Older Cenozoic dates (~35–34 Ma) came from distinct cores with zircon mantles that produced dates from ~33–30 Ma. The youngest dates, ~30–25 Ma, came from cathodoluminescent dark zircon rims, most notably in sample MS15-01 (Figure 4). MSWD values from LCS titanites ranged from ~1.2 to 7.2. Analyses from two of the eight samples, namely LCS-01 and 02, formed single populations (MSWD = 1.2). Titanites from these samples were concentrated by mineral separation techniques and analyzed in a 25-mm-diameter epoxy mount rather than with in vivo thin section analysis. It is possible (or likely) that mineral separation steps and grain picking biased the analyses toward a single titanite population. Little Cottonwood stock titanite grains from all eight samples produced dates from 37–27 Ma with modes at 34.5 and 31.5 Ma, and a different distribution for the zircon dates (Figure 5C). The oldest titanite dates ranged from ~37–32 Ma and came from structurally deeper western samples (MS15-03 and DR18-01, 02, and 03; Figure 5) near the Wasatch fault and from the satellite Ferguson stock. Samples from the center and structurally higher portions of the stock defined the middle and younger portions of the range from ~34−27 Ma. However, the incomplete spatial coverage of presently dated samples led us to postpone interpretation of an emplacement pattern pending collection of more spatially complete data.

Both zircon (125 analyses) and titanite (316 analyses) were analyzed in six samples from the Alta stock, including four samples from the border phases (D310b, B4, 89-I-11, and 88-I-9) and two samples from the central phase (C4 and C7). Zircons within the Alta stock range from an equant with a diameter of ~15 μm to elongate and ~100 μm in the long dimension. Uranium-lead dates from zircon in the AS border phases ranged from 35–31 Ma with a mode slightly younger than 34 Ma. Dates from zircons in the Alta central phase ranged from 34.5–32 Ma with a mode of 33.5 Ma. Modes of titanite dates from each Alta stock phase were younger than the zircon modes, and the ranges of titanite dates exceeded the ranges of zircon dates. The majority of titanite in the Alta border phases ranged from ~35–30 Ma. However, sample 88-I-9 from near the southern Alta stock contact contained titanites that spanned 35−25 Ma (excluding four analyses; Figure 4).

Titanites were also dated from a metasomatized Alta stock granitic rock (endoskarn sample 12-12-A2; 67 analyses) and from a wollastonite-bearing skarn (sample 11-1; 45 analyses) in the inner Alta contact aureole. Dates from the endoskarn sample ranged from 35–23 Ma (excluding three analyses) and had a complicated polymodal distribution (Figure 5A). The modes did not correlate with modes from sample 88-I-9 or in any of the Little Cottonwood stock distributions. The majority of titanite dates from the wollastonite skarn ranged from 37–29 Ma with modes at 34 Ma and 32 Ma and a single analysis at ~26.5 Ma. These modes did not correlate to the Alta stock zircon or titanite date modes, but did correlate both with modes of Little Cottonwood stock titanite dates and with the mode of Ferguson stock titanite dates (34 Ma; Figure 5).

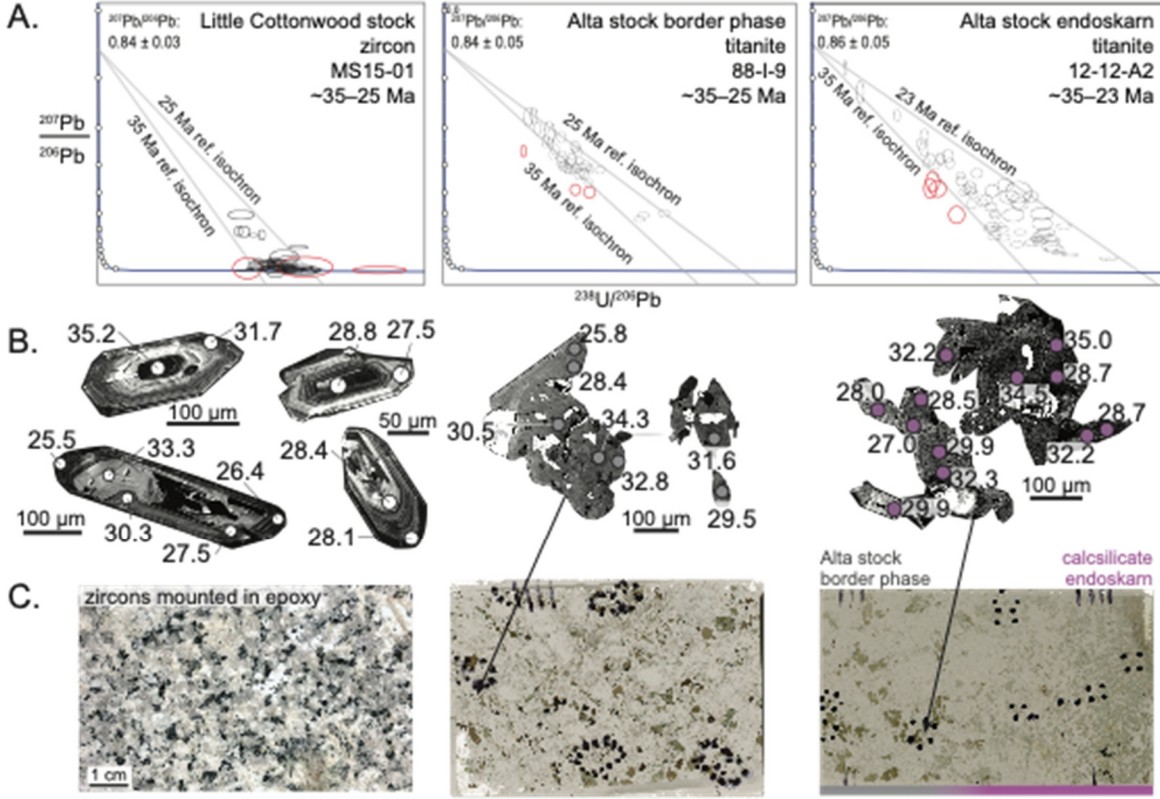

**Figure 4.** (**A**) The typical Tera–Wasserburg plots of U-Pb isotopes in zircon were nearly concordant with some outliers. The zircons in the Alta and Little Cottonwood stocks contained dates that reflected multiple generations of growth. In contrast, the Tera–Wasserburg plots of U-Pb isotopes in titanite have a large range of radiogenic and common Pb [89] and were interpreted to indicate an age range within these two samples from the Alta stock. The grey lines that bound the data were reference isochrons for the age range listed for each sample [82]. Red error ellipses were excluded based on either anomalously high [204]Pb or low Si and/or Ca concentrations, which indicated that a mineral other than titanite was sampled by the laser. (**B**) Representative cathodoluminescence (CL) and backscattered electron (BSE) images of zircon and titanite grains show the zoning and range of spot dates (colored by lithologic unit) observed in the Little Cottonwood stock (white data) and Alta stock (grey data) border phase and endoskarn samples. Laser sampling locations are labeled with the [207]Pb-corrected date and show that neocrystallized and reaction-rim grains were typically younger than recrystallized titanites. (**C**) Hand sample and petrographic thin section photographs show the unaltered macroscopic appearance of MS15-01, 88-I-9, and the gradational replacement of igneous rock by endoskarn (light purple data) in sample 12-12-A2.

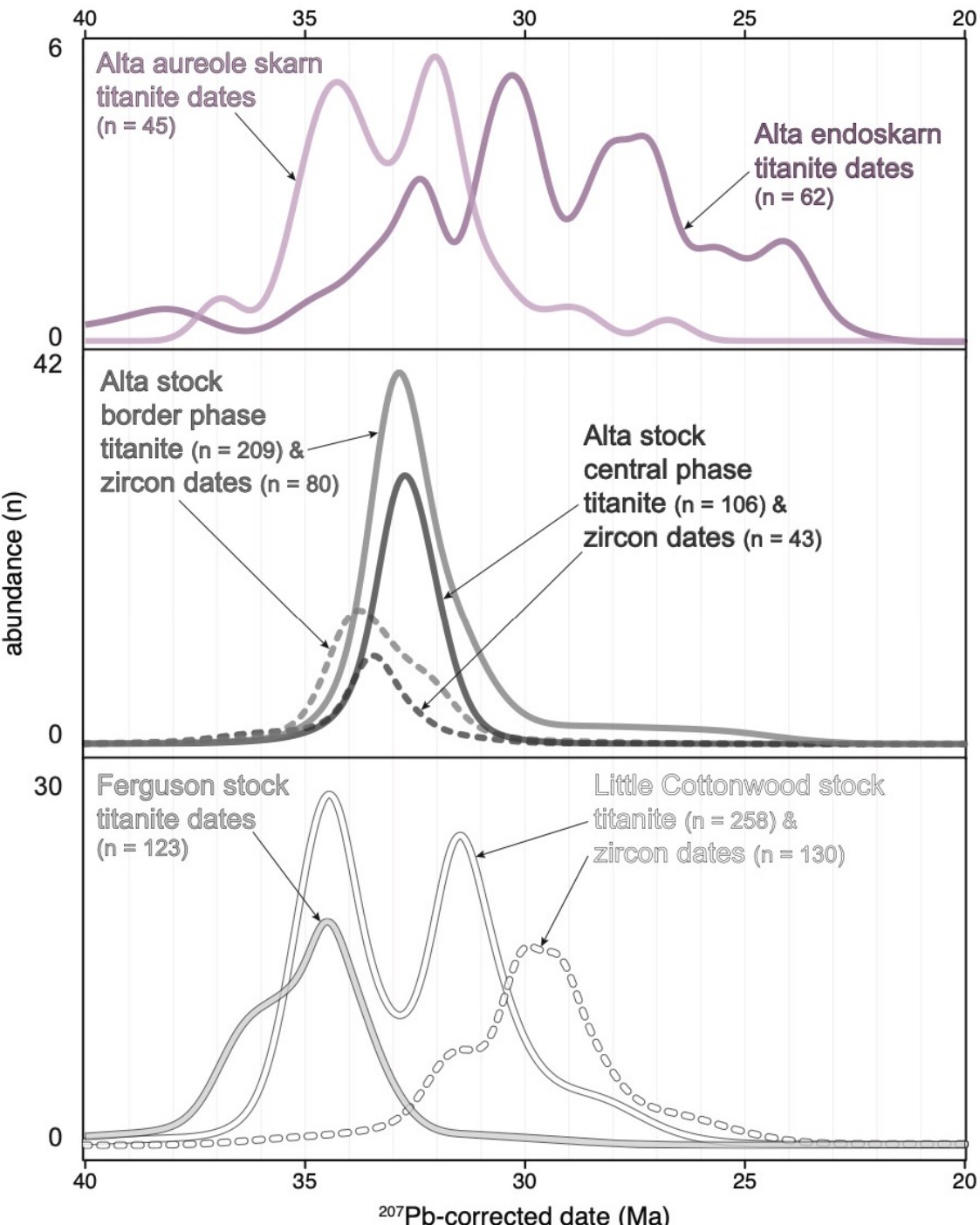

**Figure 5.** Kernel density estimates (KDEs) colored by lithologic unit of 40–20 Ma [207]Pb-corrected dates that were calculated and plotted using IsoplotR [81] for the (**top**) Alta aureole and endoskarn titanite dates (solid purple), (**middle**) Alta stock border and central phases titanite (dashed dark greys) and zircon dates (solid dark greys), and (**bottom**) Little Cottonwood stock titanites and zircon (solid white and light grey) and Ferguson stock titanite dates (dashed white and light grey). The heights of the KDE curves were normalized per grouping and the maximum abundances are labeled on the y-axis for each curve.

### 3.2.2. Trace Elements and Thermometry

The concentrations of all rare earth elements (REEs) in the Alta–Little Cottonwood titanites decreased with time (Figure 6A, Figure S1 and Table S1). Ytterbium, for example, decreased from

>$10^3$ times chondrite prior to 30 Ma to <$10^2$ times chondrite after 30 Ma. The oldest titanites defined a narrow range of REE contents with a slightly higher light (L)REE and lower heavy (H)REE contents. The europium anomaly (Eu/Eu*) in titanites ranged from 0.3 to 3, excluding outliers, with a mode of ~0.9. Pre-35-Ma titanites generally fell below 1.0, while 35–30-Ma titanites defined the full range of Eu/Eu* from 0.1–3. Analyses younger than 30 Ma also generally had an Eu/Eu* of less than 1.0. Many analyses of zircon had REE concentrations near or below the detection limit during LASS analysis and the spider plot does not include concentrations below 1× chondrite (Figure 6B). All zircon analyses showed a typical pattern of low LREE and middle (M)REE concentrations and increasing HREE content with a pronounced cerium anomaly (Ce/Ce*). Zircons increased in all REE with time, most notably Pr through Dy. The samarium content (Figure 6B) increased from <$10^2$ times chondrite prior to 30 Ma to generally >$10^2$ after 30 Ma. The Eu/Eu* (~0.1–7) in zircon became more scattered through time. Cerium anomalies (Ce/Ce*) in the Alta–Little Cottonwood zircons ranged from $10^1$–$10^4$ and were positively correlated with Eu/Eu* ($p < 0.01$; n = 22), meaning zircons with a higher Ce/Ce* tended to have a less negative Eu/Eu* (closer to 1.0). The $Ce^{4+}$ and $Eu^{3+}$ ions were both favored at higher oxygen fugacity [90,91], which is suggestive of trends in the Alta–Little Cottonwood magma oxidation state, but there was no statistically significant relationship between either anomaly and the [207]Pb-corrected date.

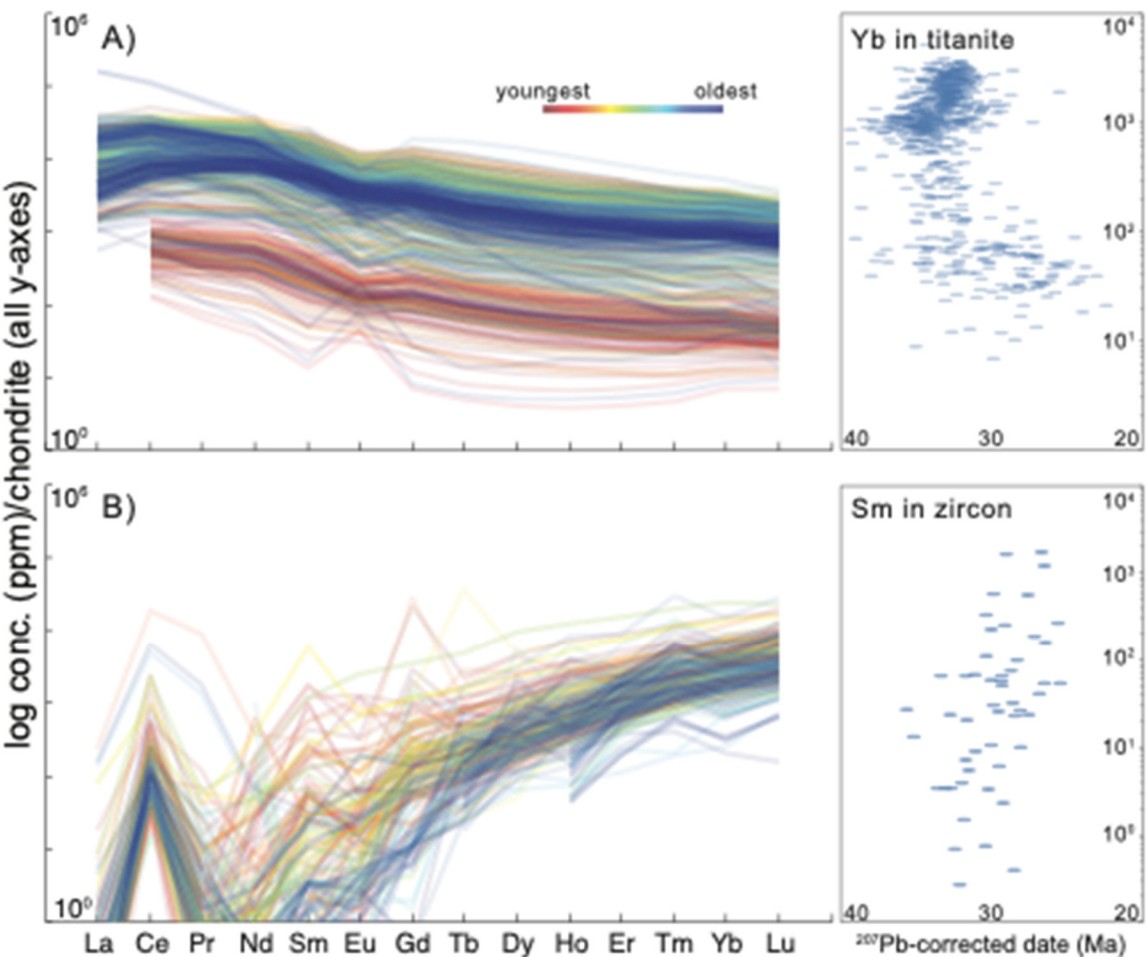

**Figure 6.** Semi-logarithmic chondrite normalized spider diagrams for all (**A**) titanite and (**B**) zircon analyses, colored using the [207]Pb-corrected dates. Older dates are blue and younger dates are red. Titanite analyses became depleted in lanthanides through time, while zircons became enriched through time. See the Data Repository File for the concentration and normalized data.

The titanium contents of Little Cottonwood stock zircons ranged from 3.8–10 ppm (Figure 7) with a dominant mode at 4.8 ppm. Zircons from the Alta border phase ranged from 2.7–8.4 ppm titanium with a mode at 4.0 ppm. The Ti contents of Alta central phase zircons ranged from 0–8.3 ppm with modes at ~3.0 and ~4.5 ppm Ti. Abnormally high Ti contents (>>10 ppm) were likely caused by the ablation of a Ti-rich phase like ilmenite, rutile, or titanite included in zircon. Inclusions were avoided during spot placement and none were observed during analysis, but mineral inclusions were observed during optical and SEM microscopy.

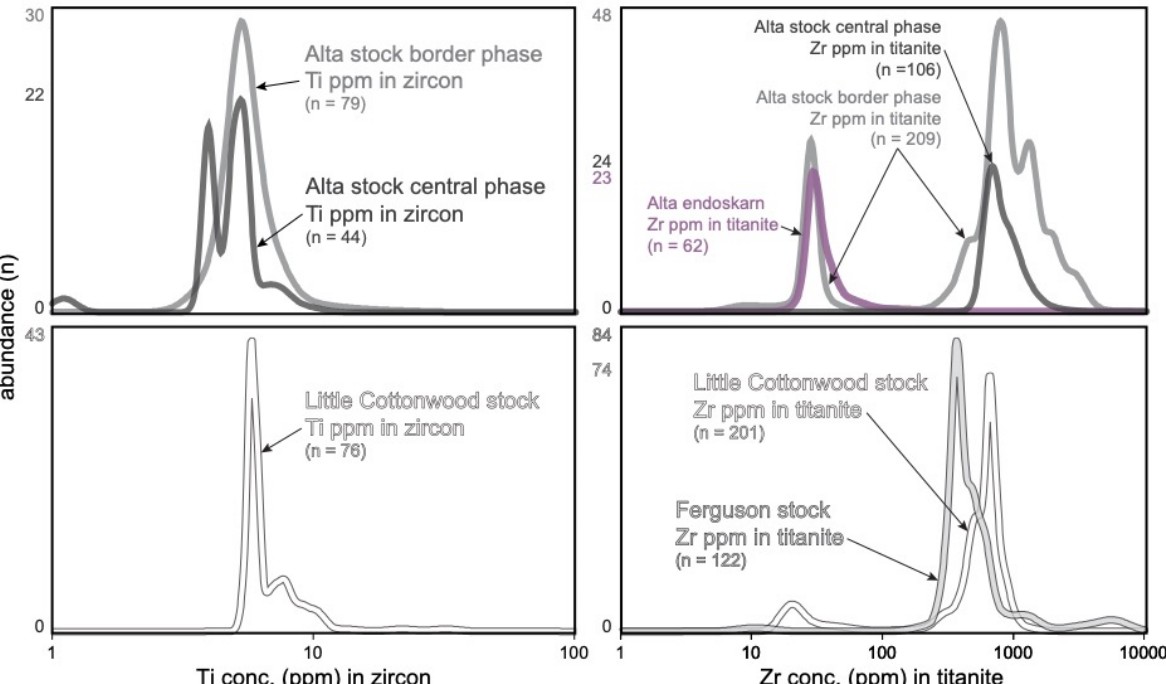

**Figure 7.** Kernel density estimates (KDEs) of the Ti content of zircon (left column) and Zr content in titanite (right column) colored by lithologic unit and calculated and plotted using IsoplotR [81] for the Alta endoskarn (purple), Alta stock border and central phases (dark greys), Little Cottonwood stock (white), and Ferguson stock (light grey). The heights of the KDE curves are normalized per grouping and the maximum abundances are labeled on the y-axis for each curve.

Zirconium contents of Little Cottonwood stock titanites ranged from 10 to >1000 ppm. The two units had different distributions. Ferguson stock titanites contained less Zr with a mode at ~400 ppm, while the Little Cottonwood stock titanites had a small mode at ~15 ppm and a large mode at ~700 ppm. Zirconium contents of the Alta stock border phase titanites ranged from ~10 to ~2000 ppm with a complex distribution. The two modes at ~15 ppm and ~750 ppm roughly corresponded to the Little Cottonwood stock titanite modes. The lower Zr mode in the Alta stock border phase titanites was found only in sample 88-I-9 from the southern margin of the pluton. The Alta stock central phase titanites had a more restricted range of Zr content from ~500–1500 ppm with a mode at ~700 ppm. The titanites from the Alta endoskarn sample 12-12-A2 contained 20–110 ppm Zr and largely overlapped with the mode defined by sample 88-I-9. Titanites from the wollastonite skarn contained anomalously high Zr contents (>>1000 ppm) that corresponded with sector zoning in the crystals. These titanites likely incorporated a non-equilibrium amount of Zr and other trace elements [92,93] and were not considered in the thermometry discussion below.

## 4. Discussion

### 4.1. Thermal History

Transforming the concentrations of tetravalent cations (Figure 7) into apparent temperatures and plotting against $^{207}$Pb-corrected dates yielded a temperature–time path for the Alta–Little Cottonwood system (Figure 8). Ti-in-zircon thermometry produced temperatures of ~650–850 ± 40 °C (using $\alpha_{TiO2}$ = 0.5 ± 0.1 and $\alpha_{SiO2}$ = 0.8 ± 0.1) with a mode of ~718 °C for both intrusions (MSWD = 1.8). This mode was consistent with the zircon saturation temperature (~725 ± 25 °C), the likely solidus (675–725 °C [94,95]) for these rocks, and the petrographically determined crystallization sequence. Titanium-in-zircon apparent temperatures from the Alta–Little Cottonwood system are hotter than results from other felsic and intermediate rocks [62]. Other studies [62,96,97] have reported Ti-in-zircon apparent temperatures lower than that predicted by Ti-in-quartz, Zr saturation, and $\delta^{18}$O quartz-magnetite pair thermometers, as well as temperatures of zircon paragenesis from phase relationships. The inconsistency reported in other studies and general sources of uncertainty could reflect: (1) misestimation (likely overestimation) of the $TiO_2$ and $SiO_2$ activities during zircon crystallization, (2) the pressure dependence of the thermometer, (3) the subsolidus open system behavior of zircon, and/or (4) prolonged Zr saturation in the magma and real low-temperature crystallization of zircons. Multiple indicators of the $TiO_2$ activity (see the Materials and Methods Section), including the absence of rutile from all samples, suggest that the $TiO_2$ activity of 0.5 ± 0.1 was reasonable, reflects the phase assemblage, and encompasses the mode ($\alpha_{TiO2}$ ~ 0.4) and 50% of the $\alpha_{TiO2}$ of experimental granitic melts [97]. $TiO_2$ activity may be lower (i.e., further from rutile saturation) during earlier crystallization of zircon and plagioclase in the magma [96], and it is unlikely that it is much higher than 0.6–0.7 at temperatures closer to the solidus. As previously noted in the Materials and Methods Section, lowering or raising the $TiO_2$ activity to a minimum of 0.3 or maximum of 0.9 changed the temperature by ±50 °C. This range of $TiO_2$ activity is possible over the entire span of zircon crystallization in magmas, but a $TiO_2$ activity near 1.0 is unlikely. Thus, a propagating uncertainty of ±0.1 (~±40 °C) encompassed nearly the entire range of temperature variation caused by any inaccuracy resulting from the $\alpha_{TiO2}$. Schiller and Finger [97] suggest that the Ferry and Watson [83] Ti-in-zircon thermometer should not be applied to $TiO_2$ undersaturated rocks, such as the I-type Alta and Little Cottonwood stocks, since it underestimates temperatures for these magmas despite the $\alpha_{TiO2}$ term in Ferry and Watson's [83] calibration. Schiller and Finger [97] further suggest a temperature-dependent correction for the $\alpha_{TiO2}$ based on rhyolite-MELTS [86] or a general +70 °C upward correction of Ti-in-zircon temperatures (Figure 8) to account for the temperature underestimate. An ad hoc +70 °C correction raised the average Ti-in-zircon temperatures from the Alta and Little Cottonwood stocks above the range of $T_{Zr}$ temperatures (≥25 °C) and the mean of the Zr-in-titanite population interpreted as magmatic (see below). Although a stronger pressure dependence of 100 °C/GPa [87] has been suggested, the pressure variations possible in this shallow system would only serve to raise apparent temperatures an additional ~5–10 °C above the $T_{Zr}$ and mean Zr-in-titanite apparent temperatures (see the Data Repository File). Several observations further corroborated the Ti-in-zircon temperatures and suggest that zircons were not altered or were otherwise open systems following initial crystallization, including the low common-Pb content of most zircons, the near concordant and concordant nature of most of the U-Pb analyses, the dates that were consistent with the inclusion of zircon in titanite, the consistent intragrain isotopic dates and crystal growth zoning, and the oscillatory zoning as opposed to patchy or sector zoning. Persistent zirconium saturation is possible based on the bulk rock chemistry, agreement between the Ti-in-zircon and the Zr saturation thermometers, and the inherited zircons in both the Alta and Little Cottonwood stocks. Conversely, the resorbed cores and mantles in some zircons suggest that Zr saturation did not persist the entire time during the crystallization of the stocks. We conclude that accounting for a lower, non-unity $\alpha_{TiO2}$ in the original calibration [83] and propagating the uncertainty on the activity terms

sufficiently reconciles the thermometry with likely zircon crystallization temperatures [96,97] and that the Ti-in-zircon apparent temperatures are likely accurate within the reported uncertainty.

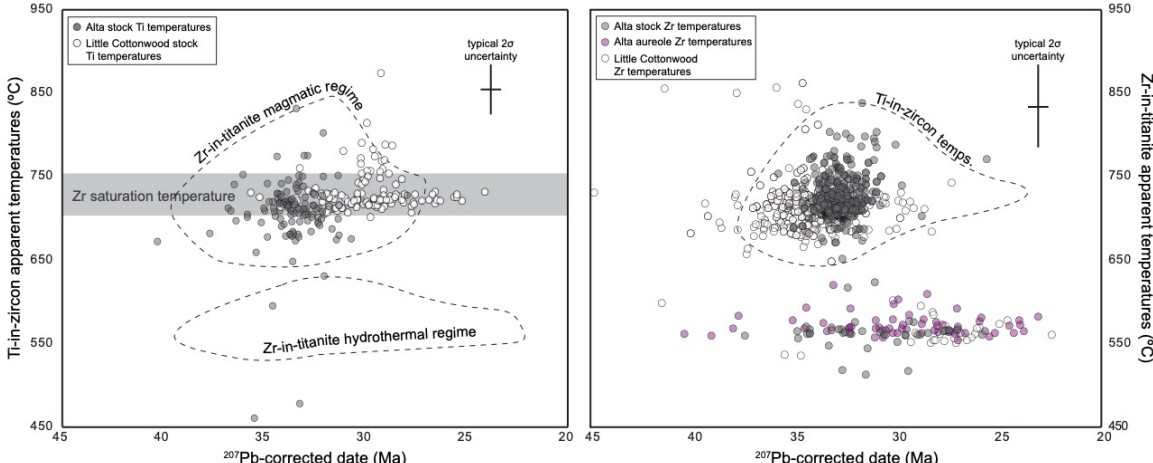

**Figure 8.** Plots of ${}^{207}$Pb-corrected dates versus Ti-in-zircon (**left**) and Zr-in-titanite (**right**) apparent temperatures calculated separately for each sample (see the Data Repository File for thermometry inputs) and colored by lithologic unit (Alta = dark grey, Little Cottonwood = white, Alta endoskarn = purple). The crosses (+) are the data transformed +70 °C, as suggested by Schiller and Finger [97], to account for a low $\alpha_{TiO2}$. The typical uncertainty bars are shown for each method but have been omitted from individual analyses for clarity. The Ti-in-zircon apparent temperatures calculated by Ferry and Watson [83] defined a unimodal population below the predicted zircon saturation temperature of ~725 °C ($T_{Zr}$; grey bar), while the Zr-in-titanite apparent temperatures defined a bimodal population that largely did not overlap the Ti thermometry data.

The Zr-in-titanite apparent temperatures formed two groups: (1) grains from both the Alta and Little Cottonwood stocks that recorded apparent temperatures ranging from ~650–800 °C with a mode at ~725 °C and were interpreted to have grown from a silicate melt, and (2) grains that recorded ~575 ± 50 °C conditions and were interpreted to have (re)crystallized in the presence of hydrothermal fluids in the solid state. These populations of Zr temperatures were consistent with the multiple textural populations of titanite observed in the Little Cottonwood and Alta stocks. The Zr-in-titanite thermometer is susceptible to similar sources of uncertainty as for the Ti-in-zircon thermometer discussed above, with a much stronger dependence on pressure. For the reasons previously outlined in the Materials and Methods and Discussion Sections, we interpreted the Zr-in-titanite temperatures as being accurate within the reported uncertainty. Samples included in the second group were LCS-02, 88-I-9, and 12-12-A2. The ranges of titanite dates from the two groups overlapped but the colder population was skewed slightly younger than the hotter population. This relationship was interpreted to record simultaneous titanite crystallization from silicate melt and fluid-mediated (re)crystallization of titanite in different parts of the system, with titanite (re)crystallization continuing after crystallization of the magmatic titanite had ceased.

### 4.2. Hydrothermal Permeability Structure through Time

The permeability structure surrounding intrusions fundamentally controls the process of fluid-infiltration-driven contact metamorphism [3,33,98]. The lack of pervasive hydrothermal titanite (re)crystallization in either the majority of the border phase or any of the central phase Alta stock samples suggests there was scarce infiltration of the hydrothermal fluids flowing through the abundant fractures and veins into unfractured volumes of the Alta stock [99] into the bulk of the non-fractured AS. Samples 88-I-9 and 12-12-A2 (Figure 4) were located at or proximal to the Alta stock wall-rock contact (0–40 m) and a recorded ≥11 Myr of titanite (re)crystallization, which is much longer than calcsilicate skarn sample 11-1 from the inner aureole (~8 Myr; ~37–29 Ma). These data suggest that the locus of

hydrothermal infiltration migrated through time (Figure 9), and further suggests that infiltration in the Alta metamorphic aureole (for example, sample 11-1) and the calcsilicate and endoskarns adjacent to the Alta stock (sample 12-12-A2) reflected different and/or multiple stages of fluid infiltration during emplacement of Alta–Little Cottonwood magmas. A diachronous and perhaps stochastic process of infiltration may reflect a feedback between prograde mineral reactions and skarn mineralization, which both exploit pore space, and the permeability structure of the host rocks [33,100].

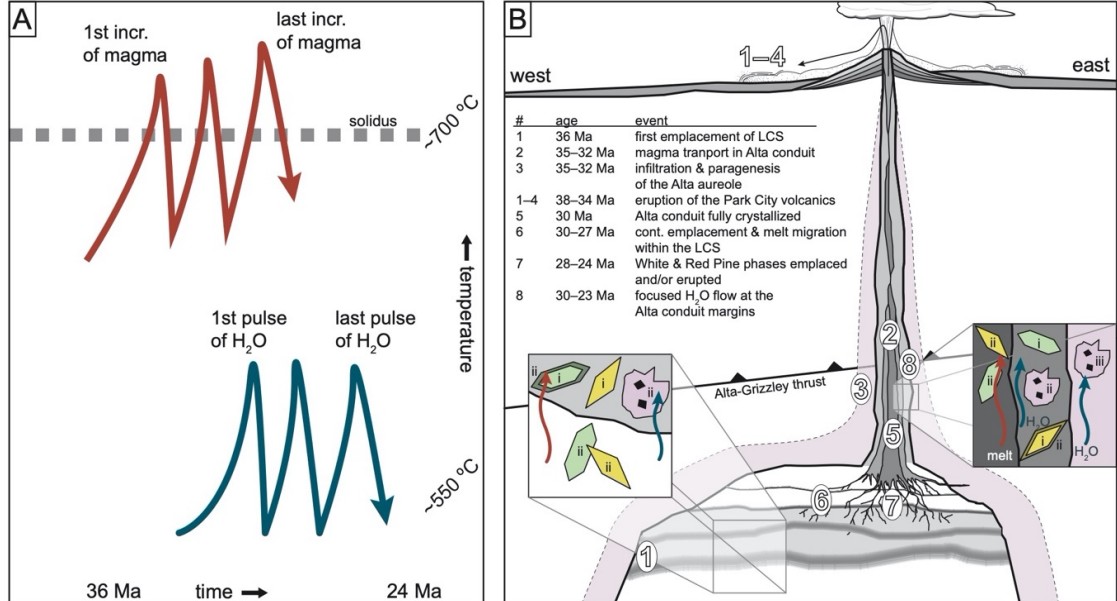

**Figure 9.** (**A**) Interpretation of the process behind the simultaneous (re)crystallization of titanite in both magmatic and hydrothermal processes suggested by the bimodal Zr-in-titanite thermometry. (**B**) A summary schematic illustrating the sequence of geologic events in the Alta stock–Little Cottonwood stock (AS–LCS) system that illustrates how incremental magmatism and pulsed hydrothermal activity could produce multiple populations (oldest = i, youngest = iii) of (re)crystallized titanite and zircon. The unit colors in (B) (dark grey = Alta, light grey = Ferguson stock, white = Little Cottonwood stock, and purple = Alta aureole) match Figures 1, 2, 4, 5, 7 and 8. The lithologic contacts and spatial extents of different units are diagrammatic and are meant to represent processes such as incremental magma emplacement, magma transport in a conduit, episodic hydrothermal infiltration, and protracted volcanic activity at the paleosurface.

## 4.3. Magma Accumulation vs. Eruptive Discharge

The total eruptive volumes of volcanic rocks related to the Alta and Little Cottonwood stocks are likely $\geq 10^3$ km$^3$ based on a minimum thickness of ~500 m [24] and a minimum outcrop area of ~1600 km$^2$ [101]. The areal exposure of the genetically related intrusive rocks is ~225 km$^2$, but the thicknesses of the intrusions are unknown. If the intrusions are assumed to be ~5–10 km thick [102], then the volume of the intrusive rocks (~1–2 × 10$^3$ km$^3$) is the same order of magnitude as the volume of erupted material. If this is the case, then the ~11 Myr intrusive duration implies a pluton growth rate that is 2–5× slower than the eruptive discharge from the system, which lasted only ~4.5 Myr (~36.5 to 32 Ma [23,24]). These rates are based on assumptions of the physical dimensions just presented and these estimates have large uncertainties and are therefore speculative. However, these very preliminary discharge estimates are consistent with other better-characterized systems [103,104] that suggest that magma discharge is a first-order control on the eruptibility of magma batches.

## 5. Conclusions

Petrochronology data indicate that pluton growth began in the structurally deepest parts of the Little Cottonwood stock at ~36 Ma. This roughly corresponds to the onset of volcanism [23,24,105] (Figures 9 and 10). From 36.5 to 31 Ma, most, if not all, of the Keetley/East Traverse volcanic sequence accumulated and the currently exposed level of the Alta stock grew, which we thus interpret to represent a conduit for magma to the earth's surface. The end of volcanism and the growth of the AS at ~31 Ma also correspond to the bulk of metamorphic titanite dates from the inner Alta aureole. However, the Little Cottonwood stock continued to grow until ~25 Ma, i.e., for several million years after the cessation of volcanism that exploited the Alta stock as a magma conduit.

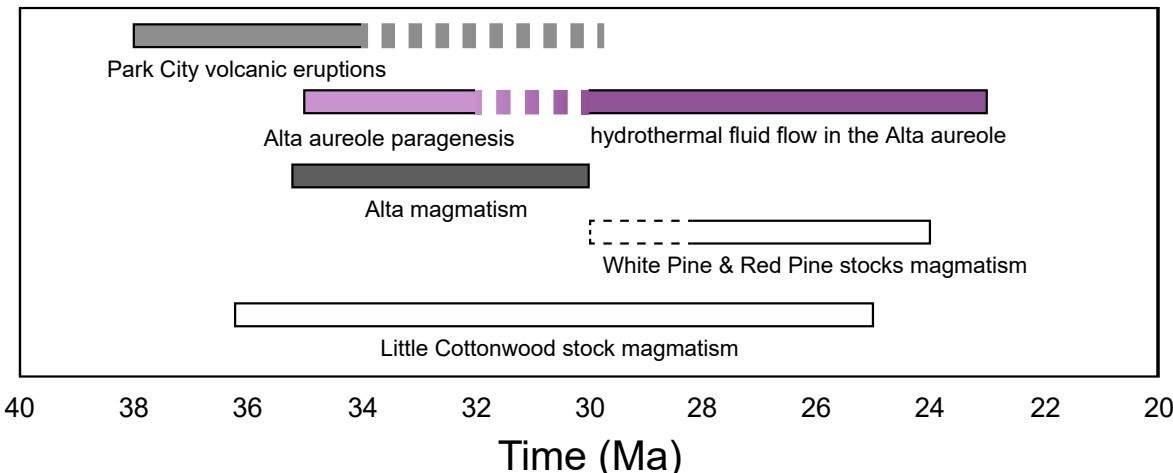

**Figure 10.** Timeline for the Alta–Little Cottonwood system based on preliminary U-Pb zircon and titanite petrochonology data. Sparse data (e.g., Park City volcanic units) and a lack of both zircon and titanite dates (e.g., Little Cottonwood stock) limits the interpretation and contributes significant uncertainty to the plotted durations. Defining the timing of these events is an ongoing research focus. The existing data also suggest these systems are episodic rather than continuous in their activity.

The isotopic dates and apparent temperatures defined a single population of zircon and two populations of titanite. The single population of apparent temperatures calculated by Ferry and Watson [83] Ti-in-zircon calibration were both consistent with the estimated $T_{Zr}$ of the magma and the magmatic Zr-in-titanite apparent temperature population. The higher temperature population of titanites was interpreted to reflect crystallization from a silicate melt and the lower temperature population of titanites was interpreted to reflect growth, likely in the solid state, in the presence of hydrothermal fluid, possibly of magmatic origin; this is consistent with the range of titanite morphologies present in the rocks. The lower-temperature titanite group had a somewhat younger maximum age but the two populations overlapped in age by ~4 Myr. We thus interpreted the titanite dates to record temporally overlapping magmatic and hydrothermal regimes. However, the low-temperature population persisted to a much younger minimum age, indicating that hydrothermal activity outlasted crystallization of the Alta stock by several million years. Titanites from the southern margin of the Alta stock recorded hydrothermal (re)crystallization through this duration, suggesting the presence of a narrow hydrothermal conduit in the AS adjacent to its southern contact.

In summary, these relations suggest that the Alta stock served as a conduit for magma to the surface from 36−31 Ma, then as a conduit for hydrothermal fluids for another 6−8 Myr. After the magma conduit shut off at ~31 Ma, the Little Cottonwood stock continued to grow until at least 25 Ma and perhaps as late as 23 Ma.

The spatial coverage of present dates from the Little Cottonwood stock is insufficient to infer an emplacement pattern. We speculate that the youngest dates from the Little Cottonwood stock,

which come from its structurally highest portions, may reflect upward melt migration and/or remobilization from younger, related intrusions.

**Supplementary Materials:** The following are available online at http://www.mdpi.com/2076-3263/10/4/129/s1, Figure S1: U-Pb Plots, Table S1: Petrochronology Data.

**Author Contributions:** Conceptualization: M.A.S., J.M.B., and J.R.B.; Data curation: M.A.S., C.W.F., and N.D.U.; Funding acquisition: M.A.S., J.M.B., and J.R.B.; Investigation: M.A.S., J.M.B., J.R.B., C.W.F., C.J.B., D.D.R., S.J.C., and N.D.U.; Methodology: M.A.S.; Visualization: M.A.S.; Writing—original draft: M.A.S., J.M.B., and J.R.B.; Writing—review and editing: M.A.S., J.M.B., J.R.B., C.W.F., C.J.B., D.D.R., S.J.C., and N.D.U. All authors have read and agreed to the published version of the manuscript.

**Funding:** This research was funded by the National Science Foundation, grant number 1853496.

**Acknowledgments:** We would like to acknowledge Allen Glazner, Drew Coleman, and Lukas Baumgartner for important discussions during the preparation of this manuscript. We would like to acknowledge Andrew Kylander-Clark and Diego Fernandez for assistance with the isotopic analysis, and thank the two reviewers for their constructive and helpful comments.

**Conflicts of Interest:** The authors declare no conflict of interest. The funders had no role in the design of the study; in the collection, analyses, or interpretation of data; in the writing of the manuscript, or in the decision to publish the results.

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
