# Peer review of "Simultaneous Magmatic and Hydrothermal Regimes in Alta–Little Cottonwood Stocks, Utah, USA, Recorded Using Multiphase U-Pb Petrochronology"

_geosciences, doi:10.3390/geosciences10040129_

Round 1

Reviewer 1 Report

Dear All

I read with pleasure the paper by Stearns et al. The paper presents new preliminary data regarding U-Pb and trace elements petrochronology from both zircon and titanite from Alta-Little Cottonwood stocks in Utah.

The presented geochronological data lead the Authors to present an alternative model where intrusion and hidrothermal activities play an important role. The methodology is very well explained and it is the most precise method at the moment.

I think that the paper is worth to be published and I require just few minor revisions, mainly regarding the figures and few things through the text.

I suggest to better explain the microstructural relations among the different populations of titanite.

I also suggest not to use so many acronymous if you do not have problem with the length of the text

Regarding the figure, Figure 1 has some major problems.

you need at first to add a legend in both maps (a and b)

In the cross section you must add a vertical scale and you heve to explain which is the meaning of the dashed lines

You need also to add some mesurements of strike and dip of the dikes in figure b, to support your geological cross section. I suggest also to add isotherms in map b.

In Figre 3 it is not clear  to which unit do you refer, please explain better and put also the name of the samples that you located in fifure 1b and c.

In Figure 4 rewrite caption b as the first image is about zircons and not titanite

In Figure 5 please explain what is the dashed line in the lowermost diagram

Hoping this could help

Reviewer 2 Report

This is an excellent paper, well written paper and the work should be published.  The authors have provided an elegant approach to explain a controversial problem, the origin of the Wasatch Intrusive Belt.  I only have a few comments, all but one involve reference bookkeeping. These are summarized below.

Lines 389-413:  The authors discuss the lower than expected Ti in zircon and propose three possible reasons for this.  All three possibilities are eliminated in subsequent text and no explanation is given for the low Ti. This is not revisited in the conclusions.  Either this should be explained or it should be stated explicitly that the cause of these low values is as yet undetermined.

Line 162-163:  (Kylander-Clark et al., 2013; 2017).  The 2017 references is Kylander-Clark 2017 (no et al.)

Line 210: Wiedenbeck et al., 1995; 2007;  Do you mean 2004?  The second Wiedenbeck reference in the reference list is 2004.

Line 389:  Piwinskii, 1968; Naney, 1983). Neither of these citations are in the reference list.

Line 425: Gerdes et al., 1995;  Not in reference list

Lines 622-623:  Kohn, M.J., 2017 is not cited in text.

Line 736:  Wiedenbeck, M., et al., 2004,  List authors in reference.
